# Fibrous PVA Matrix Containing Strontium-Substituted Hydroxyapatite Nanoparticles from Golden Apple Snail (*Pomacea canaliculata* L.) Shells for Bone Tissue Engineering

**DOI:** 10.3390/bioengineering10070844

**Published:** 2023-07-17

**Authors:** Aldi Herbanu, Ika Dewi Ana, Retno Ardhani, Widowati Siswomihardjo

**Affiliations:** 1Doctoral Study Program, Faculty of Dentistry, Universitas Gadjah Mada, Yogyakarta 55281, Indonesia; aldiherbanu@mail.ugm.ac.id (A.H.); widowati@ugm.ac.id (W.S.); 2Department of Dental Biomedical Sciences, Faculty of Dentistry, Universitas Gadjah Mada, Yogyakarta 55281, Indonesia; retnoardhani@mail.ugm.ac.id; 3Research Collaboration Center for Biomedical Scaffolds, National Research and Innovation Agency of the Republic of Indonesia, Yogyakarta 55281, Indonesia; 4Department of Dental Biomaterials, Faculty of Dentistry, Universitas Gadjah Mada, Yogyakarta 55281, Indonesia

**Keywords:** bone tissue engineering, cell viability, fibrous matrix, PVA, strontium-substituted hydroxyapatite, golden apple snail (*Pomecea canaliculata* L.) shells

## Abstract

A scaffold that replicates the physicochemical composition of bone at the nanoscale level is a promising replacement for conventional bone grafts such as autograft, allograft, or xenograft. However, its creation is still a major challenge in bone tissue engineering. The fabrication of a fibrous PVA-HA/Sr matrix made of strontium (Sr)-substituted hydroxyapatite from the shell of *Pomecea canaliculate* L. (golden apple snail) is reported in this work. Since the fabrication of HAp from biogenic resources such as the shell of golden apple snail (GASs) should be conducted at very high temperature and results in high crystalline HAp, Sr substitution to Ca was applied to reduce crystallinity during HAp synthesis. The resulted HAp and HA/Sr nanoparticles were then combined with PVA to create fibrous PVA-HAp or PVA-HA/Sr matrices in 2 or 4 mol % Sr ions substitution by electrospinning. The nanofiber diameter increased gradually by the addition of HAp, HA/Sr 2 mol %, and HA/Sr 4 mol %, respectively, into PVA. The percentage of the swelling ratio increased and reached the maximum value in PVA-HA/Sr-4 mol %, as well as in its protein adsorption. Furthermore, the matrices with HAp or HA/Sr incorporation exhibited good bioactivity, increased cell viability and proliferation. Therefore, the fibrous matrices generated in this study are considered potential candidates for bone tissue engineering scaffolds. Further in vivo studies become an urgency to valorize these results into real clinical application.

## 1. Introduction

In bone tissue engineering, the use of a designated scaffold which enables cell migration, adhesion, and proliferation is an alternative to traditional bone transplant therapy. Bone hydroxyapatite (HAp), the most common and well-known apatite species utilized in orthopedics surgery, is one among many substances that is used in bone regenerative surgery which closely resembles bone mineral [1]. The extracellular matrix (ECM) that makes up bone interacts with inorganic (carbonated apatite mineral) and organic (mostly type-1 collagen) components to form a fibrous structure at the nanoscale level [2]. Using an electrospinning method, which creates a fibrous structure with fiber diameters varying from nanometer to micrometer, this nanoscale structure can be recreated in a synthetic scaffold. The fabricated fibrous structure can be adjusted to mimic the native ECM in bone in terms of structure, porosity, and surface area [3,4].

Using electrospinning, numerous scaffolds or synthetic ECMs can be created, from polymer, ceramic, metallic, and composite nanofibers [4]. Several polymer/bioceramic composites have been used widely to create the fibrous scaffold to replicate the structure of bone at the nanoscale level. Overall, when compared to scaffolds made from organic materials alone, scaffolds made from composites exhibit better properties and performances, such as higher mechanical strength, better cellular proliferation, greater bioactivity, and stronger hydrophilicity [5,6,7,8,9,10,11,12,13]. 

Regarding inorganic components in bones, synthetic hydroxyapatite (HAp) is known to be naturally compatible with bone tissues and closely matches the mineral found in human tooth and bone. Thus, HAp becomes an especially appealing material for bone and tooth implants [14,15,16]. A lot of investigations have shown that HAp ceramics do not exhibit toxicity, inflammation, or pyrogenetic reaction. Besides, extensive research has been being conducted to provide synthetic HAp from various resources such as fish scales, bones, eggshells, and corals. Although the conversions of Ca-containing biogenic resources into HAp have been found to be successful, the high crystallinity of the resulted HAp still becomes challenging, due to the decreased osteoclast activities which retarded the bone remodelling process. Therefore, it is necessary to improve the biological properties of the resulted HAp from biogenic resources by transforming HAp into low-crystalline ones so that the degradation rate of HAp can be adjusted to support the ideal bone remodelling process. As previously described in other studies, extra ions addition, such as Magnesium (Mg) or Strontium (Sr), into HAp structure can reduce crystallinity due to lattice parameter alteration when the ions substitute calcium [17,18,19,20,21]. 

Furthermore, even though HAp can bind directly to the host bone, it has great fibrous tissue growth between the implant and bone [14]. Besides, native bone tissue is best described as a nanohybrid consisting of ∼30 wt % collagen (a protein-based hydrogel template) reinforced by a finely dispersed mineral phase (70 wt %) consisting of nanosized apatite crystals. The organic components of bone provide flexibility and resilience, while the inorganic components support bone in its hardness and rigidity [22]. To mimic bone tissue unique properties, it is assumed that the optimal synthetic bone tissue equivalent will also consist of an organized and dense polymeric matrix containing dispersed HAp nanoparticles of low crystallinity. Despite extensive studies which have been conducted related to HAp, only a few researchers have investigated the performance of HAp in the form of hybrid polymeric matrix for bone tissue engineering. In view of that, in this study it was considered that poly(vinyl alcohol) is one of the potential polymers for tissue engineering which can be utilized to create fibrous structure in a hybrid polymeric matrix with HAp. Poly(vinyl alcohol) or PVA is a water-soluble poly-hydroxy polymer and semi-crystalline which is usually used in the textile, paper, and healthcare sector industries due to its good film-forming, thus it can facilitate the electrospinning process. It is also biodegradable and compatible with practically all bodily tissues [23,24,25]. 

In the efforts to imitate the native ECM for bone tissue engineering, the current study investigated and reported the fabrication of HAp from the golden apple snail (*Pomacea canaliculate* L.) by lowering its crystallinity with Sr ions substitution. Upon obtaining the apatite (with or without the Sr ions substitution), the inclusion of HAp and strontium-substituted hydroxyapatite (HA/Sr) which are similar to biological bone material was conducted consecutively, to result in fibrous PVA/HAp and PVA/HA/Sr matrices. A hydrothermal technique was used to synthesize HAp and HA/Sr, which was then mechanically combined with a PVA/HAp and PVA/HA/Sr solution before being electrospun to create a fibrous PVA/HAp and PVA/HA/Sr hybrid matrices. In this work, the physicochemical and nano-microstructural characteristics of the matrices were examined. The scaffold performance was also evaluated in terms of bioactivity, swelling behavior, protein adsorption, cell viability, and cell proliferation. 

## 2. Materials and Methods

### 2.1. Materials

The reagents for synthesizing HAp or HA/Sr from the shells of golden apple snail (*Pomacea canaliculata* L.) were prepared, i.e., nitric acid (HNO_3_) with >98% purity (Merck, Darmstadt, Germany), ammonium dihydrogen phosphate (NH_4_)2HPO_4_) with >98% purity (Merck, Darmstadt, Germany), ammonium hydroxide (NH_4_OH) with >98% purity (Merck, Darmstadt, Germany), and strontium nitrate (Sr(NO_3_)_2_) with purity level >98% (Sigma Aldrich, Baden-Württemberg, Germany). The source of calcium (CaO) was obtained from golden apple snail (*Pomacea canaliculata* L.) shells (GASS). The HNO_3_ was used to increase solubility, NH_4_OH for pH control or OH group provider, and (NH_4_)2HPO_4_) for phosphate group sources. 

### 2.2. Methods

#### 2.2.1. Preparation of Nanoscale HAp and HA/Sr from GASS

Upon being biologically identified to ensure homogeneity, GASS of *Pomacea canaliculata* L. were cleaned using distilled water, boiled for 1 h, oven-dried at 40 °C for 24 h, ground, and sieved until they became a powder of ±100 mesh. The <100 mesh sieved GASS powder was calcinated at 900 °C for 4 h to convert it into the CaO phase. The CaO phase was added into 100 mL of distilled water and stirred to form a suspension and added with 2M nitric acid (HNO_3_). Stirring was continued at a speed of 600–700 rpm with 90 °C temperature at a pH of 10, resulting in Ca(NO_3_)_2_ solution. 

Next, diammonium hydrogen phosphate (ADP) or (NH_4_)_2_HPO_4_ was added and mixed with Ca(NO_3_)_2_ solution to form HAp by phosphate precipitation. Wise dripping of NH_4_OH at a speed of 1 mL/min using a burette was conducted to keep the solution at a pH of 9–10. Maturation was done consecutively by continuous stirring for 24 h until HAp precipitates were formed. The HAp precipitates obtained were sieved, washed with distilled water, and dried at 100 °C to obtain HAp powder. The product was then calcined at 1200 °C in a conventional air furnace (Nabertherm, Lilienthal, Germany). The resulting specimen was encoded as “HAp”. 

To prepare HA/Sr, the doping process or addition of Sr ions into HAp was carried out as follows. At first, Sr(NO_3_)_2_ solution was added to the Ca(NO_3_)_2_ solution to achieve a concentration of Sr equivalent to 2 and 4 mol % with a constant (Ca+Sr)/P final ratio of 1.67. Then, the phosphate mixture of (NH_4_)_2_HPO_4_, Ca(NO_3_)_2_, and Sr(NO_3_)_2_ solutions was adjusted to be a pH of 10 by adding NH_4_OH solution. Following that, the maturation process was carried out at 90 °C using a magnetic stirrer for 24 h at room temperature. The precipitate supernatant and Sr-substituted HAp were directly separated by vacuum filtration without a washing procedure. The Sr-doped HA slurry was filtered, oven-dried, and calcined at 1200 °C in a conventional air furnace (Nabertherm, Lilienthal, Germany). The resulting specimens were encoded “HA/Sr” containing either 2 or 4 mol % strontium.

#### 2.2.2. Preparation of Spinning Solution

The HA/Sr powder of 2%, i.e., 0.12 g, was dissolved in 50 mL of distilled water for 1 h, then PVA weighing 6 g was added to the HA/Sr solution to produce a composition of 12% (*w*/*v*) with constant stirring at 90 °C for 3 h. A ratio of 85:15 (*v*/*v*) was used to combine the PVA solution and HA/Sr solution based on the preliminary experiments, and referred to in a previous study [26]. To study the impact of HA/Sr incorporation into the matrix, the amount of HA/Sr powder addition was adjusted at 0, 2, and 4 mol % of the total mass of the polymer in solution. The stirring process was maintained for the next 24 h at room temperature for reaction maturation. The solution that has undergone maturation would then be used in the electrospinning process for membrane manufacturing.

#### 2.2.3. Fabrication of Fibrous Matrices

Before the electrospinning process, the solution was sonicated (stirred with ultrasonic waves) for 1 h to obtain a homogeneous, well-dispersed solution, without any air bubbles in the solution. The solution was put into a 10 mL plastic syringe equipped with a needle of 0.5 mm hole diameter. A voltage of 15 kV was applied to the tip of the needle. The solution from the syringe was exposed by electrospinning onto aluminum foil on an aluminum plate placed at 12 cm from the syringe tip. The polymer feed rate of the solution was maintained at 0.25 mL/h. The experiments were conducted at room temperature for PVA-HA, PVA-HA/Sr-2 mol %, and PVA-HA/Sr-4 mol %. In general, the preparation of fibrous PVA-HA/Sr matrices was as described in Figure 1. 

#### 2.2.4. Morphological Analysis of Fibrous Matrices

The microstructure of the PVA, PVA-HAp, PVA-HA/Sr-2 mol %, and PVA-HA/Sr-4 mol % matrices and the morphology of the nanofiber in the matrices were analyzed by Scanning Electron Microscope (SEM, JEOL-JSM-6510LA, Tokyo, Japan). The average diameter of the fibers was calculated based on the measurement of 100 randomly selected fibers. The morphology and size of HAp and HA/Sr particles dispersed in the PVA matrix were determined by transmission electron microscopy (TEM, HT7700, Hitachi, Tokyo, Japan), operating at 200 kV.

#### 2.2.5. Crystallography Analysis of Fibrous Matrices

The crystallographic properties of PVA, PVA-HAp, PVA-HA/Sr-2 mol %, and PVA-HA/Sr-4 mol % matrices were determined by X-ray diffractometer (Powder XRD, AXS D8 Advance Eco, Bruker GmbH, Karlsruhe, Germany) at the range 2*θ* of 10–60° using Cu–Kα radiation at the wavelength (λ) = 0.154 nm.

#### 2.2.6. Infrared Spectra of Fibrous Matrices 

The FTIR (Thermo Scientific Nicolet iS10, Tokyo, Japan) was used to measure the functional groups within the PVA, PVA-HAp, and PVA-HA/Sr-2 mol %, PVA-HA/Sr-4 mol % matrices. In short, the ground sample to be analyzed was mixed with potassium bromide (KBr) to emit infrared light that would be read on the screen monitor. The FTIR equipment is operated in the wavelength range of 400–4000 cm^−1^.

#### 2.2.7. Swelling Ratio of Fibrous Matrices

The method used to measure swelling behavior was referred to Patriati et al. [21]. In brief, matrices were cut into pieces of about 10 × 10 mm^2^, then the initial weight (*W*0) was measured. The cut matrices were then soaked in distilled water for 24 h. Samples were washed, filtered with filter paper to remove water impurities on the surface of the matrices, and then weighed in a wet state (*Wt*). The swelling ratio at 24 h measurement was repeated 3 times (*n* = 3) in each group. The calculation of the swelling ratio was performed with the following Equation (1).
(1)$$Swelling\;Ratio\;\left(\%\right)=\left(\frac{Wt-W0}{W0}\right)\times 100\%$$

#### 2.2.8. Protein Adsorption onto the Surface of the Fibrous Matrices

The method used to measure protein adsorption was referred to in previous studies by Januariyasa et al. [27] and Ana et al. [28]. Initially, the matrix was cut into rectangles with a sample mass of 1 g each. The samples were soaked in 70% ethanol for 1 h and then washed with PBS 3 times. Consecutively, the samples were incubated in 1 mL of PBS containing 10% FBS for 1 h at 36.5 °C. After incubation, soaking of the samples was continued in 1 mL of PBS for 15 min to remove proteins that were not completely attached to the matrix. The procedure was repeated 3 times, then the washing solution was added into the FBS solution from the remaining incubation. Finally, the remaining protein in the solution was measured by UV-vis spectrophotometer (UV-1800, Shimadzu, Kyoto, Japan), and the amount of protein adsorbed onto the matrix surface was measured by subtracting the protein concentration after incubation from protein concentration before incubation. 

#### 2.2.9. Degradability of Fibrous Matrices

For the test of matrix degradability, a method applied by Patriati et al. [21] was also referred to. At first, dried matrices were placed in PBS (750 µL) at 37 °C with pH 7.4. The PBS supernatant was taken out and subsequently exchanged with fresh PBS at 1, 3, and 24 h. The amount of PVA in each solution was measured from the supernatant by the colorimetric method. The absorbance of a solution at 280 nm was measured with a UV-Vis spectrophotometer (UV-1800, Shimadzu, Japan). Accelerated PVA degradation measurement with high acidic solution 1N HCl was not conducted in this study.

#### 2.2.10. Bioactivity in SBF and Biocompatibility of Fibrous Matrices in MC3T3-E1

To examine bioactivity of the fibrous matrices, bioactivity test was conducted in simulated body fluid (SBF). Simulated body fluid (SBF) was prepared to mimic the physicochemical condition of native body fluid. Furthermore, biocompatibility examination of fibrous PVA, PVA-HAp, PVA-HA/Sr-2 mol %, and PVA-HA/Sr-4 mol % cell studies using osteoblast-like cells MC3T3-E1 were performed. 

##### Bioactivity of Fibrous Matrices in SBF

The matrices were cut into 20 × 20 mm^2^ rectangles and immersed in SBF 1 × solution, as described by Kokubo and Takadama [29], at pH 7.4, with the set temperature of 36.5 °C. After 3, 7, or 10 days of immersion, the samples were washed with distilled water and dried at ambient temperature. Mineralization of the samples was determined by SEM (JEOL, JSM-6510LA, Tokyo, Japan), and the composition of the apatite formed on the fibers was determined by EDS. 

##### Cell Culture

Mouse osteoblast cells (MC3T3-E1) were cultured in α-MEM medium (Gibco, Waltham, MA, USA) supplemented with 10% FBS (Gibco, Waltham, MA, USA), Penicillin–Streptomycin (Gibco, Waltham, MA, USA), and Fungizone 0.5% (Gibco, Waltham, MA, USA). Prior to cell seeding, all matrices of 10 × 10 mm^2^ were sterilized using ethylene oxide gas (EOG) in a dry environment and using a low-temperature method. The matrices were laid flat on the bottom of the well with a fixation from a sterilized plastic ring. The cells were seeded onto matrices at a density of 5 × 10^4^ cells/well in 24-well plates. The incubation was done at 37 °C in 5% CO_2_ flow for 24 and 48 h.

##### Cell Viability Assay with MTT

Cell viability was studied by MTT assay for each period of incubation (24 and 48 h). The measurement was conducted in triplicate for each type of matrices and the well without matrix as a control. In general, upon the incubation-reached predeterminant period, the medium was discarded. An MTT reagent (Biobasic, Amherst, NY, USA) with a concentration of 0.5 mg/mL was added to the well in 250 μL volume and incubated for 4 h. Then, dimethyl sulfoxide (DMSO, Merck KGaA, Darmstadt, Germany) was added to the well at 250 μL/well. The absorbance (abs) was recorded by Tecan Spark^®^ (Tecan Trading AG, Zurich, Switzerland) at 570 nm. The cell viability was calculated by the following Equation (2):(2)$$Cellviability\left(\%\right)=\frac{Absorbance\;of\;matrix-Absorbance\;of\;control\;medium}{\left(Absorbance\;of\;control-Absorbance\;of\;control\;medium\right)}\times \;100\%$$

##### Cell Adhesion and Proliferation Observation

A membrane from each type of matrices was cultured in 24-well plates for 48 h for cell morphology observation. After incubation, the matrices were washed using PBS to discard cells that were not attached to the matrices, then the matrices were incubated in a 2.5% glutaraldehyde solution for 2 h at 4 °C to fixate the cells. The membranes were then dehydrated by soaking in various graded ethanol solutions (20, 30, 40, 50, 60, 70, and 100%) and followed by drying for overnight. The membranes were ready for SEM observation to examine adhesion and proliferation of the cells on the matrices.

#### 2.2.11. Statistical Analysis

If the data were normally distributed and variation between groups was homogenous, statistical analysis was conducted with analysis of variance (ANOVA), either a one-way or two-way, followed by the LSD (Least Significant Difference) test. If the data were not normally distributed and/or the variation between groups was not homogeneous, then nonparametric analysis was chosen for the analysis. A *p*-value less than 0.05 was considered statistically significant. 

## 3. Results

A series of characterizations for the PVA, PVA-HAp, PVA-HA/Sr-2 mol %, and PVA-HA/Sr-4 mol % matrices have been conducted. The results are reported in this section. 

### 3.1. Morphology and Microstructure of Fibrous Matrices

From the micrograph of the matrices (Figure 2), a two-dimensional (2D) fibrous matrices of the PVA, PVA-HAp, PVA-HA/Sr-2 mol %, and PVA-HA/Sr-4 mol % formed nonwoven fiber matrices. A nonwoven fiber matrix formed was close to the ECM structure in bone tissue. Thus, the method was confirmed as a proper method to develop a scaffold as a synthetic ECM. There were no white beads appearance in the microstructure of PVA (Figure 2A) without any HAp or HA/Sr. The Ca and P were found scattered or distributed along the fibrous matrix, demonstrated by white circle beads in PVA-HAp and PVA-HA/Sr (Figure 2B–D). It was shown that the white beads in PVA-HA/Sr-4 mol % were in the >1 micrometer size, showing agglomeration (Figure 2D). From the SEM micrograph, the fiber diameter was calculated, and the results are described in Figure 3. The smallest diameter was found in PVA without any HAp or HA/Sr. The fiber diameter was in the nanoscale and ranged from 360.50 ± 77.16 to 505.51 ± 78.60 nm in the PVA with the incorporation of either HAp or HA/Sr-2 mol % and HA/Sr-4 mol %. The incorporation of HAp or HA/Sr was found to increase fiber diameter. More strontium (Sr) ion substitution resulted in bigger fiber diameters. In general, PVA forms intermolecular hydrogen bonding with HA, which increases spinnability. The lower the crystal size, the higher the spinnability, which resulted in larger fiber diameter.

From the SEM pictures taken in the study, the crystal size of the HAp, HA/Sr in either 2 mol % or 4 mol % can be calculated. The crystal size of the HAp, HA/Sr-2 mol %, and HA/Sr-4 mol % was 37.74, 28.82, and 20.52 nm, respectively.

From the TEM analysis as shown in Figure 4, the appearance of the particles in nanoscale size was detected. The PVA particle was a smooth circle in 200 nm. Agglomerated particles with raw surfaces in the 50 nm size were found in the PVA-HAp matrix. Irregular flakes in the form of bars or oval with a particle size of 500 nm and 100 nm were found in PVA-HA/Sr-2 mol % and PVA-HA/Sr-4 mol %, respectively. 

### 3.2. Crystallography

As shown in Figure 5A, the diffraction pattern of PVA without HAp or HA/Sr content has one broad peak around 2*θ* degree of 19.5, which corresponds to the PVA blend. The crystal signature of apatite was found in around 25.5, 31.0, and 32.5 with the PVA in 20.4 2*θ* degree in the matrices with HAp or HA/Sr. The diffraction pattern of 2% and 4% PVA-Ha/Sr showed that the peaks with the addition of Sr ions decreased in number and intensity relative to the PVA-HAp peaks. Meanwhile, additional peaks at around 25.84, 31.74, and 32.87 in PVA-HA/Sr-2 mol % as well as peaks at 25.80, 31.67, 32.80 in PVA-HA/Sr-4 mol % appeared along with the decrease in Sr ions substitution. It shows that the more the concentration of Sr ions, the less the HAp crystals are present in the nanofiber structure. The peak intensity of PVA-HAp, PVA-HA/Sr-2 mol %, and PVA-HA/Sr-4 mol % was found to be in semi-crystalline phases.

### 3.3. FTIR Spectra of Fibrous Matrices

The FT-IR spectra of PVA-HAp and PVA-HA/Sr fibers showed the increasing peaks from the bonds owned by PVA, HAp, and HA-Sr, which were dominated by PVA. The peak at wave number 919 cm^−1^ is the distinct characteristic of apatite, which is related to the stretching vibration of PO_4_^3−^. As can also be seen from Figure 5B, the intensity of the peak at wave number 1095 cm^−1^ increased when HAp was added. The PO_4_^3−^ peak was seen to be related to the C-O-C peak in all fibrous matrices. The PO_4_^3−^ peak in the 600 cm^−1^ range owned by PVA decreased when HAp or HA/Sr was added. These results indicated that PO_4_^3−^ bonds from the PVA decreased when HAp or HA/Sr was added, which indirectly means that the factual presence of apatite in the fibrous matrices structure also increased.

### 3.4. Swelling Ratio, Degradability, and Surface Protein Adsorption

It was observed from the analysis of the swelling rate in one time point of 24 h after incubation (Figure 6) that PVA-HA/Sr-4 mol % absorbed the largest amount of water (2.2 ± 1.2%) followed by PVA-HA/Sr 2 mol % (1.5 ± 0.8%), PVA-HAp (1.3 ± 07%), and PVA (0. ± 0.5%). The rate of water absorption among different fibrous matrices was similar to the rate of protein sorption on the surface of the matrices, with the surface of PVA-HA/Sr-4 mol % having the highest capability to adsorb protein (0.60 ± 0.003%), followed by PVA-HA/Sr 2 mol % (0.47 ± 0.003%), PVA-HAp (0.40 ± 0.002%), and PVA (0.30 ± 0.00%).

Regarding the rate of the degradability of the fibrous matrices, it was found that the rate was the largest in PVA-HAp at 3 h incubation, followed by PVA-HA/Sr-4 mol % and PVA-HA/Sr-2 mol % at the same 3 h incubation time. The degradability rate of PVA without any HAp or HA/Sr was consistently the lowest at 1, 3, and 24 h. In the early period of incubation, i.e., 1 and 3 h, the rate of the PVA-HAp was the highest. However, when it reached 24 h incubation, the rate from the highest to the lowest was observed in PVA-HA/Sr-4 mol %, PVA-HA/Sr-2 mol %, PVA-HAp, and PVA, as depicted in Figure 7. Cumulatively for 24 h incubation, the total degradation of PVA-HA/Sr-4 mol % was less than 50% and found to be the lowest when compared to PVA-HAp and PVA-HA/Sr-2 mol %.

### 3.5. Bioactivity of Fibrous Matrices

As shown in Figure 8, there was no indication of apatite formation on the surface of the PVA fibrous matrix even after 10 days of immersion in SBF. For the PVA-HAp, on day 3, mineralized apatite has already appeared clearly. There were protrusions along the fiber, which showed agglomerations of HAp particles. Mineralized apatite was clearly observed on day 7 with the apatite minerals formed along the fiber, indicated by a large-size white dashed circle. The apatite tends to cluster rather than spread evenly along the fiber. For the PVA-HA/Sr-2 mol %, apatite precipitation was seen on day 7 after immersion. Apatite minerals were clearly observed to be in precipitated form along the fiber of PVA-HA/Sr-4 mol %. There were no obvious differences in the amount and size of apatite growing on the fibrous matrix surfaces from day 3 to 10 among PVA-HAp, PVA-HA/Sr-2 mol %, and PVA-HA/Sr-4 mol %.

### 3.6. Cell Viability, Adhesion, and Proliferation towards Fibrous Matrices

The results of the cell viability study (Figure 9) demonstrated good biocompatibility of all fibrous matrices. Substitution of Ca with Sr at 2–4 mol % was found to be safe, friendly, and did not do harm to the MC3T3-E1 cells. The PVA nanofiber has a lower percentage of cell viability than other samples, while 2% PVA-HA/Sr nanofiber has a greater percentage of cell viability or allows cells to grow better. However, the PVA-HA/Sr-4 mol % fibrous matrix decreased more percentage of cell viability levels than the PVA-HA/Sr-2 mol % matrix.

The SEM demonstrated the behavior of MC3T3-E1 on the surface of the fibrous matrices. After 48 h of incubation, cells were observed to adhere onto the surface of all matrices. A micrograph of PVA-HAp or PVA-HA/Sr exhibited firm MC3T3-E1 adherence clearly. In the PVA-HA/Sr-2 mol %, cell proliferation was detected, while in the PVA-HA/Sr-4 mol %, extension of cell cytoplasm was observed. Figure 10 shows the SEM picture of the MC3T3-E1 on the surface of the cells. Phenomena on adhesion, cytoplasm extension, and proliferation could be detected from the figures. 

## 4. Discussion

In this study, HAp from the shells of GAS (*Pomacea canaliculata* L.) has been successfully developed. The Sr ions substitution to reduce HAp crystallinity has also been conducted successfully, resulting in a smaller crystal size of HAp. The more the concentration of Sr ions (4 mol %), the lower the crystallinity of the apatite, ranging from 37.74, 28.82, and 20.52 nm for HAp, HA/Sr-2 mol %, and HA/Sr-4 mol %, respectively. A comprehensive characterization of the HAp and HA/Sr have been reported separately. Consecutively, a composition of 12% (*w*/*v*) hybrid solution containing PVA alone or in combination with HAp or HA/Sr was prepared. In the final formulation, a ratio of 85:15 (*v*/*v*) was used to combine the PVA solution and HAp or HA/Sr. The electrospinning technique resulted in fibrous matrices of PVA-HA/Sr-4 mol %, PVA-HA/Sr-2 mol %, PVA-HAp, and PVA.

Based on the macrostructural observation, 2D fibrous matrices have been successfully fabricated. Under the SEM micrograph observation, it was confirmed that a nanoscale nonwoven fibrous structure has been fabricated to mimic a native ECM in bone in terms of the microarchitecture, porosity, and surface area [3,4]. The data also affirmed that the morphology of PVA nanofiber without HAp or HA-Sr demonstrated very minimal fiber beads compared to the morphology of PVA nanofiber with HAp or HA-Sr content in it. In addition to that, the HAp or HA/Sr were found scattered along the fiber as evidenced by the dispersed distribution of white circle beads. Despite the lowest crystal size of HA/Sr-4 mol %, the PVA-HA/Sr-4 mol % have shown an unfavorable effect on nanofiber morphology due to the appearance of bead–beads agglomeration in more than 1 micron in size. Previous study reported that beads morphology along the nanofiber decreased the ability of nanofiber to support cell proliferation on the surface of fibrous matrix when compared to the one without bead [30].

Moreover, based on the SEM morphological analysis, the fiber diameter size produced in this study is within the range of fiber size in human bone ECM, around 100–450 nm [31]. Fiber diameter size is one of the parameters that can affect cell and material interactions. A decrease in fiber diameter generally increases the specific surface area of the scaffold [30]. The resulted fibrous matrices in this study are good for materials applied in tissue engineering, because they increase the exposed surface area, thus providing a large site for protein and cell attachment and can increase ion-exchange activities between the material and its surrounding body fluids [30,32].

The results of TEM analysis demonstrated that individual particles of the fibrous matrices have dimensions in the range of 100 to 500 nm and form agglomerations with agglomeration size up to the nanometer scale and are porous and homogeneous. If it is compared to the average particle size of the matrices calculated from TEM, which was 87.5 nm, it indicated that the fibrous matrices could provide good potential to be incorporated into human bone structure with the particle size between 50 and 500 nm. Furthermore, by considering the crystal size of the minerals in bone ECM, which is around 50 to 100 nm [33], the matrices containing HAp or HA/Sr resulted in this study could support osteoblast attachment better than larger particles >100 nm [34].

The structure and morphology of PVA, PVA-Hap, and PVA-HA/Sr fibrous matrices obtained have different shapes, such as circular or spherical PVA with a smooth surface. When PVA was hybridized with HAp into PVA-HAp, the morphology and shape of fibrous particles changed dramatically from spherical to coarse particles. In the case of PVA-HA/Sr, nanofiber exhibited a relatively dense and irregular flake structure (oval shaped), with an average size varying between 100 and 500 nm. The morphology of the PVA-incorporated HAp and HA/Sr particles revealed that HAp or HA/Sr particles were uniformly distributed in the PVA fibrous matrices.

Regarding the X-ray diffraction patterns obtained from the matrix examination, a broad peak of PVA was found in 2*θ* of 19.5°, which corresponded to PVA blend. Meanwhile, according to a study by Jia et al. [35], a peak of PVA has been detected in 2*θ* around 20.4°. It can be assumed that there was intensive interaction between PVA and HAp or HA/Sr to slightly move the PVA peak. The signature of apatite has been detected clearly in the resulted XRD patterns of PVA-HAp, PVA-HA/Sr-2 mol %, and PVA-HA/Sr-4 mol %. Anyhow, the results also showed that the addition of Sr decreased the intensity of PVA-HAp peaks relatively. This can be identified from the peak (002) at 2*θ* of around 25.84°, peak (121) at 2*θ* of around 31.85°, and peak (300) at 2*θ* of around 33.01°. Meanwhile, several peaks appeared when the addition of Sr decreased namely, peaks at 2*θ* of around 25.84°, 31.74°, and 32.87° and peaks at 2*θ* of around 25.80°, 31.67°, and 32.80°. It suggested that the more the concentration of HA/Sr, the less the HAp crystals are present in the nanofiber structure, whereas the semi-crystalline phase of all the matrices may be due to the presence of HAp and HA/Sr particles in the fibrous structure, which broke the molecular chains of the polymer. The disconnection of the molecular chains resulted in an increasingly small crystalline region, thus reducing the peak intensity to semi-crystalline [35].

With respect to FTIR spectra, the stretching vibration of PO_4_^3−^ was identified with the appearance of a peak at wave number 919 cm^−1^, as a distinct signature of apatite. The intensity of the peak at wave number 1095 cm^−1^ which belongs to PVA increased when HAp or HA/Sr was added. It was also shown that the OH^−1^ peak at 3386 cm^−1^ in PVA has shifted towards a larger wave number of 3396 cm^−1^, along with the addition of HAp or HA/Sr into the PVA blend. This peak shift indicated strengthening hydrogen bonds in the fibrous structure due to the presence of apatite particles from HAp or HA/Sr that carry additional OH^−1^ functional groups [36]. The OH^−1^ functional groups belonging to HAp and HA/Sr interacted and formed hydrogen bonds with OH^−1^ functional groups of PVA [26], whereas in PVA-HA/Sr-2 mol %, the broad peak belongs to OH^−1^ shifted to a smaller wave number (3386.24 cm^−1^) than PVA-HA/Sr-4 mol %. This can be explained by the fact that the shifting was because of micron-sized agglomerations along the fiber. Agglomeration has been a sign that the number of HA/Sr particles dispersed within the fibrous structure was less than the ideal condition as in the addition of HAp. As a result, the surface area of HA/Sr particles that can interact with the polymer became smaller than that of HAp. Therefore, the opportunity for the OH^−1^ functional group of HA/Sr to interact with the OH^−1^ functional group of the polymer also became smaller [31,35,36].

In this study, after physical characterization was performed, the impact of the microstructure on the matrix capability to absorb water and adsorb protein was investigated. Hence, matrix swelling analysis at one time point of 24 h reflecting its capability to absorb water after being soaked in PBS for 24 h was observed. Swelling is the event of polymer nanofiber absorbing water into its structure when placed in a hydrolytic environment, resulting in an increase of scaffold mass [37]. In general, adding HAp or HA/Sr increases the matrix hydrophilicity, because HAp adds an extra hydroxyl group, as indicated also in the FTIR results. Consequently, it causes the scaffold to absorb more water [38]. The larger porosity of the scaffold may also have an impact on its higher swelling ratio [38,39]. The smaller nanofiber diameter that was found in this study might result in a greater surface area and pore, which might boost the capacity to absorb water.

This may possibly be the cause of the lower swelling ratio occurred at PVA-HA/Sr-4 mol % compared to PVA-HA/Sr-2 mol %. The PVA-HA/Sr-4 mol % matrix has significant micron-sized agglomerations, which lower the porosity of the scaffold and the amount of water that can permeate into pores. Overall, these findings demonstrated that the porous fibrous PVA-HA/Sr matrix can permit the penetration of bodily fluids, proteins, and cells in addition to the predicted swelling of water. Therefore, that might promote the development of new bone tissues [40].

Regarding protein adsorption, in general, the addition of HAp or HA/Sr into the PVA matrix also increased protein uptake capability. As seen from the results, a significant increase occurred from PVA, PVA-HAp, PVA-HA/Sr-2 mol %, to PVA-HA/Sr-4 mol %. It was demonstrated that the addition of HAp or HA/Sr nanoparticles to PVA has a significant effect on the overall protein uptake of the matrix. The presence of HAp or HA/Sr nanoparticles changed the surface characteristics of the matrix, by presenting additional binding sites for proteins. The presence of Ca, P, and/or Sr on the fibrous matrix surface provided additional options for the charged protein surface to electrostatically bind to Ca, P, and/or Sr on HAp or HA/Sr [41]. In more detail, inside the solutions with a pH that tends to be neutral (such as in vitro tests to mimic the pH of biological fluids in the human body), protein surfaces are generally negatively charged, so the presence of Ca and Sr on HAp or HA/Sr (in the form of positively charged ionic Ca^2+^ and Sr^2+^ has a dominant role in protein uptake [3,41]. Another factor that could increase protein uptake in this study is the increased surface area of the matrix [42].

When implanted in the body, the fibrous matrices will come into direct contact with body fluids. When in contact with liquid, the matrix will experience swelling or absorb liquid, then, until it reaches a certain point, the matrix will be degraded. The degradation process without a catalyst, i.e., in PBS, began with erosion from the surface area of the matrix, while the core matrix has not changed. The PBS could break down the molecular bonds of the matrix, started by breaking down the weak entanglements [43]. When assessing the results of the degradability test, it seemed contradictory to the theory that crystallinity did affect degradation rate, because the highest degradation at 3 h was found in PVA-HAp. Meanwhile, HAp is more crystalline than HA/Sr. However, it was shown that the trend changed after 24 h, in which fibrous matrix with lower crystalline HA/Sr-4 mol % had a higher degradability rate [43,44]. Thus, it was assumed that crystallinity affects only partially the first stages of water diffusion in the polymer matrix.

The results of the bioactivity test in SBF showed that the addition of HAp or HA/Sr into the fibrous spinning solution improved bioactivity of the matrices. This also proved that the transformation from the shells of GAS or *Pomacea canaliculata* L. to Hap- or Sr-substituted HAp occurred completely. This study also affirmed the results from previous studies that apatite precipitated at a specific site first, then the precipitation spreads on the overall fibrous surface [9,45,46]. The presence of negative ions such as OH^−^ and PO_4_^3−^, and possibly along with CO_3_^2−^ on HAp, exerted an electrostatic force on Ca^2+^ ions to form an amorphous calcium phosphate (ACP) layer that has a high Ca^2+^ content, in a way to minimize the charge difference between the material surface and the formed layer. Later, the ACP layer with high Ca^2+^ content triggered the formation of the next ACP layer with high PO_4_^3−^ content, or low Ca^2+^ content. This second ACP layer was formed to balance the charge of the first ACP layer. The third layer was apatite, which contains both Ca^2+^ and PO_4_^3−^ ions [47,48].

Upon completing physical, mechanical, and chemical characterizations, cell studies using MC3T3-E1 were conducted. It was confirmed that the PVA without any HAp or HA/Sr had the lowest percentage of cell availability. The PVA-HA/Sr-2% seemed to fasten the proliferation, shown by the percentage of cell viability, which was >100%. It was also confirmed that Sr ions substitution into HAp from *Pomacea canaliculata L* was in the safety range. The results of SEM observation showed that osteoblast cells attached to and multiplied quite well on the surface of the fibrous matrices. The MC3T3-E1 primarily grouped and developed some subconfluent structures. The more frequent clustering of the cells, which was correlated with the results of cell viability assay, was most likely a result of the presence of HAp and HA/Sr on the matrices (PVA-HAp and PVA-HA/Sr). 

In accordance with the finding, earlier research has discovered that adding ceramics to polymer-based materials can boost cell activity on the scaffold. It has been affirmed that the presence of HAp content in the matrices increased its osteoconductivity [9,10,12]. Osteoconductivity was probably resulted from the chemical characteristics of HAp, which, as the scaffold dissolves, can release ions into the environment. These ions then work as guides to draw osteoblast cells to attach to and multiply on the scaffold surface [48]. The findings on protein adsorption might potentially contribute to the improvement in cell survival, since proteins can mediate osteoblast cell adhesion to the scaffold surface. Furthermore, in this study, the resulted electrospun matrices were soft ones. Previous studies confirmed that cells easily migrate, attach, and proliferate on a soft matrix [13,49], as it has been demonstrated in this study.

Although it has been demonstrated in this study that the PVA combined with Sr ions-substituted HAp from *Pomacea canaliculata* L. resulted in good composite properties, further studies need to be conducted. In this study, bioactivity was only observed with SEM. Other bioactivity tests must be conducted further. Moreover, to confirm the safety and biofunctionality of the resulted materials, a series of biocompatibility and biofunctionality testing in osseous and nonosseous environments are also needed, both in vitro and in vivo. Meanwhile, regarding the reproducibility of this study, other indigenous biogenic resources can be synthesized into HAp with Sr ions substitution using the same methods to provide biomaterials for patients.

## 5. Conclusions

It has been validated in this study that the shells of GAS *Pomacea canaliculata* L. have been successfully transformed into HAp with the proposed procedure conducted in this study. The Sr ions substitution during the synthesis resulted in low crystalline HAp. When the HAp or HA/Sr nanoparticles were incorporated into PVA and electrospun, hybrid fibrous PVA-HAp or PVA-HA/Sr matrices with good properties can be obtained and was potential to be applied in bone tissue engineering, especially for a damaged tissue which needs thin, soft, and osteoconductive matrices. The nanofiber diameter increased by the addition of either HAp or HA/Sr nanoparticles. The percentage of the swelling ratio increased and reached the maximum value in PVA-HA/Sr-4 mol %, as well as its protein adsorption. Furthermore, the matrices with HAp or HA/Sr incorporation exhibited good bioactivity and increased cell viability and proliferation. Therefore, the fibrous matrices generated in this study are considered potential candidates for bone tissue engineering scaffolds. Further in vivo and/or 3D microfluidic bone-on-a-chip studies become an urgency to valorize these results into real clinical application.

## Figures and Tables

**Figure 1 bioengineering-10-00844-f001:**
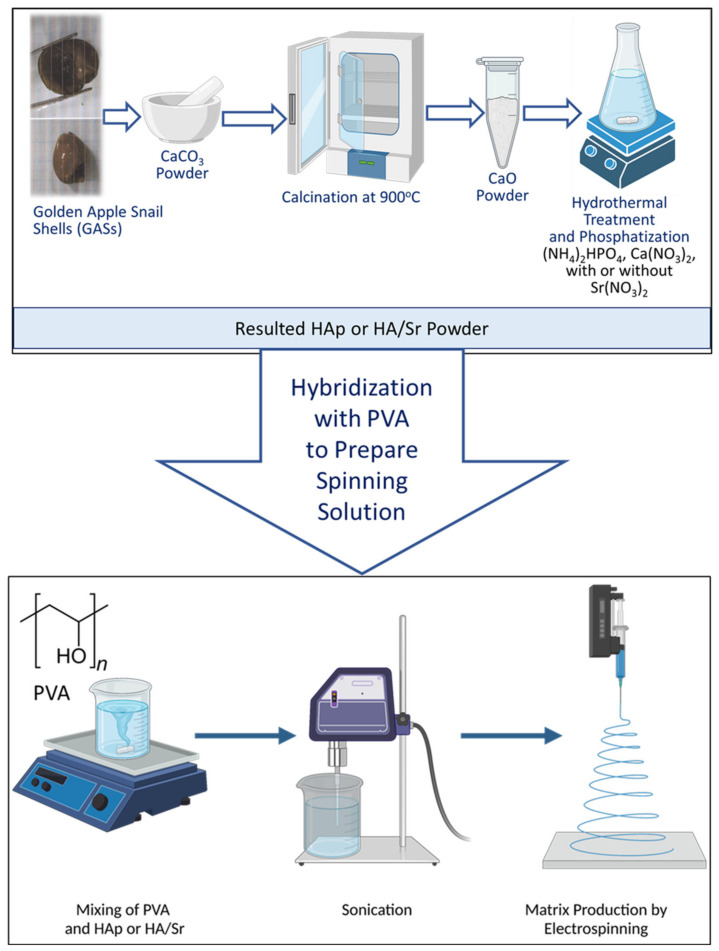
A diagrammatical picture for the PVA-HAp, PVA-HA/Sr 2 mol %, and PVA-HA/Sr 4 mol % preparation procedure. The nanoscale HAp and HA/Sr were prepared from the shells of golden apple snail (*Pomacea canaliculata* L.) through calcination and phosphatization. The PVA-HAp or PVA-HA/Sr, either 2 or 4 mol %, were prepared by mixing.

**Figure 2 bioengineering-10-00844-f002:**
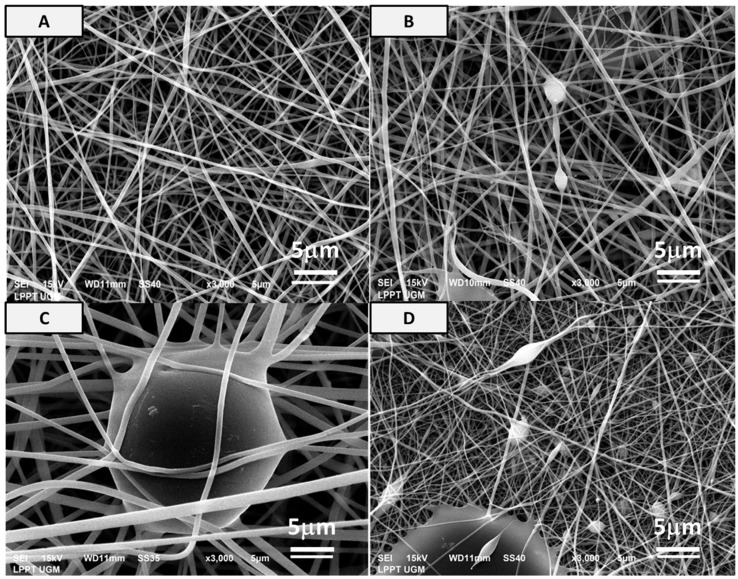
Microstructure of (**A**) PVA, (**B**) PVA-HAp, (**C**) PVA-HA/Sr-2 mol %, and (**D**) PVA-HA/Sr-4 mol % fibrous matrices at 3000× magnification. It was shown that nonwoven fabric could be formed. There was an appearance of white circle beads or white charging in SEM pictures from the PVA incorporated with either HAp or HA/Sr, which demonstrated the Ca-P agglomeration, but the size was dependent on the type of HAp or HA/Sr incorporated. The highest proportion and size of the beads were found in PVA-HA/Sr-4 mol %.

**Figure 3 bioengineering-10-00844-f003:**
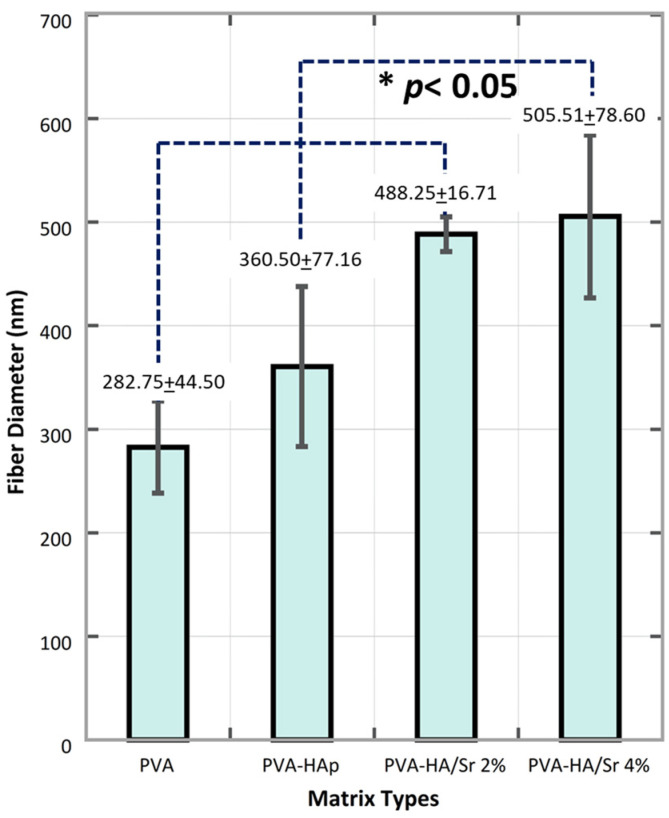
Fiber diameter was found in nanoscale size, ranged from 282.75 ± 44.50 in PVA, then 360.50 ± 77.16, 488.25 ± 16.17 to 505.51 ± 78.60 nm. The more Sr ions substitution, the larger the diameter of the fiber.

**Figure 4 bioengineering-10-00844-f004:**
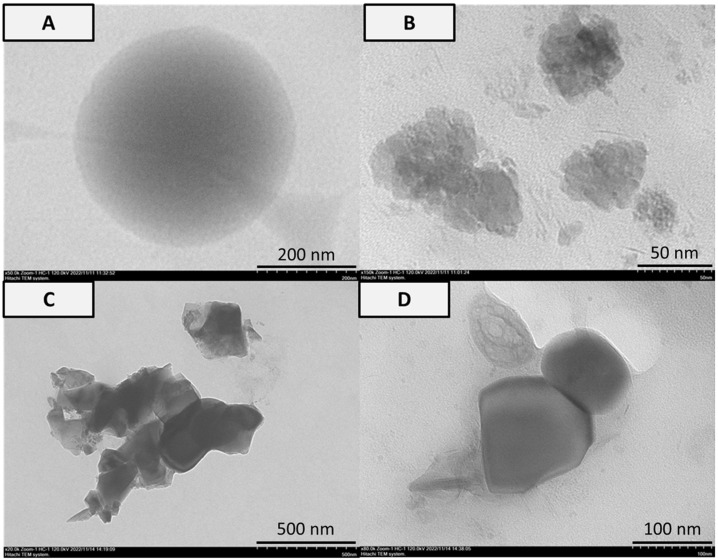
Analysis of TEM in (**A**) PVA, (**B**) PVA-HAp, (**C**) PVA-HA/Sr-2 mol %, and (**D**) PVA-HA/Sr-4 mol % fibrous matrices. Smooth circle particle in 200 nm was found in the PVA matrix. Agglomerated particles and flakes in the form of bars or oval were found in the matrices with either HAp or HA/Sr.

**Figure 5 bioengineering-10-00844-f005:**
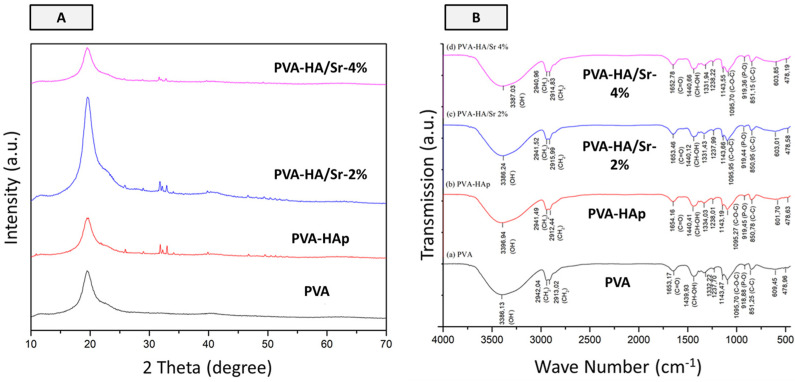
Results of X-ray diffraction analysis (**A**) and FT-IR spectra (**B**). Based on the diffraction patterns, it was found that all fibrous matrices were in semi-crystalline phases. From (**A**), it was noticed that the signature of apatite at around 25.5 and 32.5 2*θ* degrees was found in PVA-HAp, PVA-HA/Sr-2 %, and PVA-HA/Sr-4%. The FTIR spectra indicated the increase of distinct apatite peaks when HAp or HA/Sr was incorporated in fibrous matrices. In (**B**), the peak at wave number 919 cm^−1^ is the distinct characteristic of apatite, which is related to the stretching vibration of PO_4_^3−^. As can also be seen from (**B**), the intensity of the peak at wave number 1095 cm^−1^ increased when HAp was added. The PO_4_^3−^ peak was seen to be related to the C-O-C peak in all fibrous matrices. The PO_4_^3−^ peak in the 600 cm^−1^ range owned by PVA decreased when HAp or HA/Sr was added.

**Figure 6 bioengineering-10-00844-f006:**
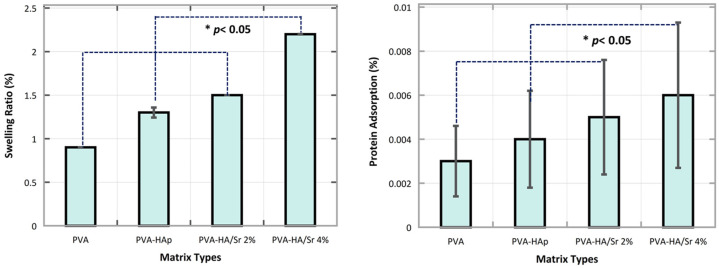
Swelling ratio at 1 time point after 24 h incubation in PBS. The rate of the swelling was the highest in PVA-HA/Sr-4 mol %. Swelling capability increased significantly by the addition of HAp with more Sr ions substitution (*p* < 0.05). Protein adsorption of the fibrous matrices had the same tendency, with the highest being also found in PVA-HA/Sr-4 mol %.

**Figure 7 bioengineering-10-00844-f007:**
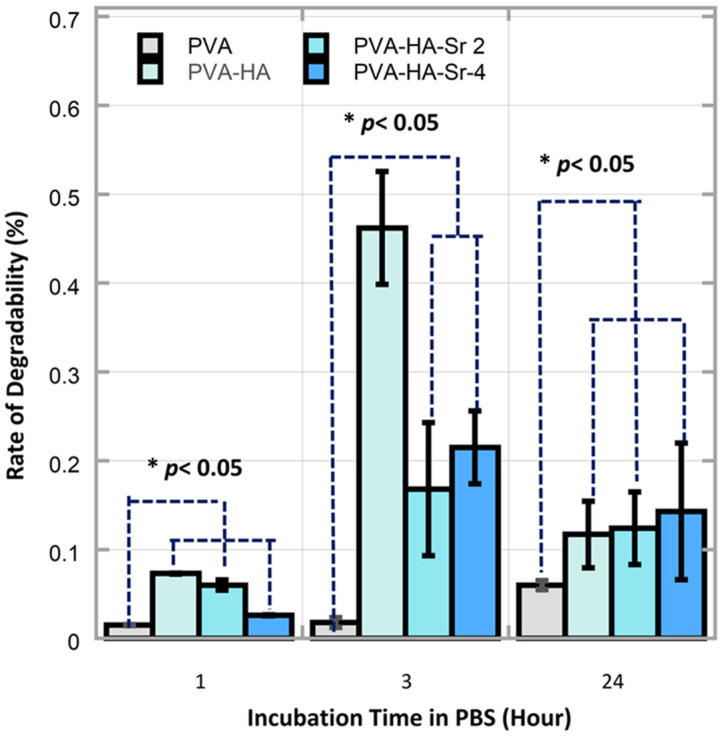
There were significant differences on the degradability rate between different fibrous matrices and different time points (*p* < 0.05). The rate of the degradability in the PVA-HAp matrices in 1 and 3 h were found to be the highest (*p* < 0.05).

**Figure 8 bioengineering-10-00844-f008:**
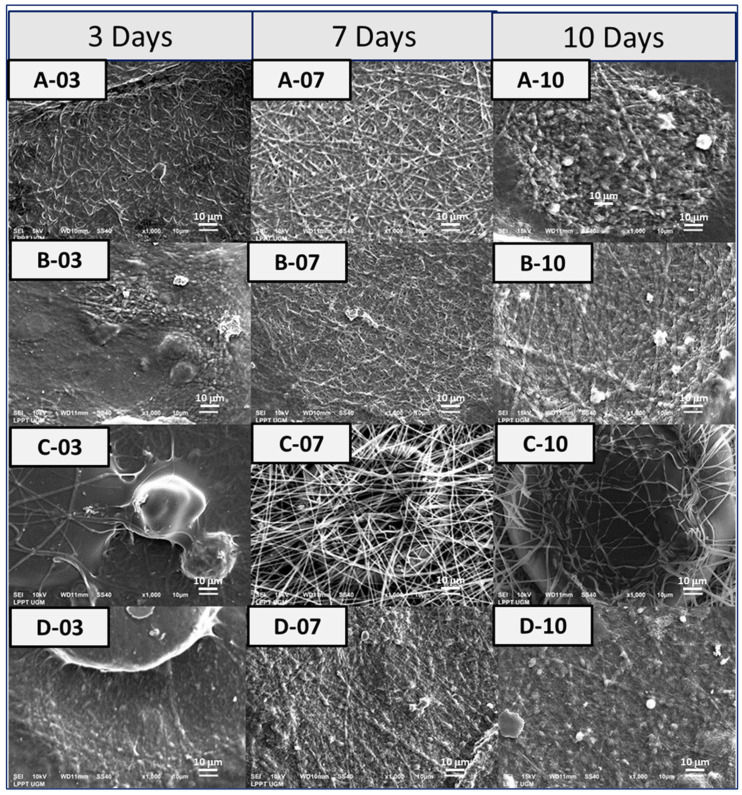
Bioactivity testing was performed in SBF [24] for (**A**) PVA, (**B**) PVA-HAp, (**C**) PVA-HA/Sr-2 mol %, and (**D**) PVA-HA/Sr-4 mol % fibrous matrices. The observation in SEM with 1000× magnification was taken after 3, 7, and 10 days’ immersion in SBF.

**Figure 9 bioengineering-10-00844-f009:**
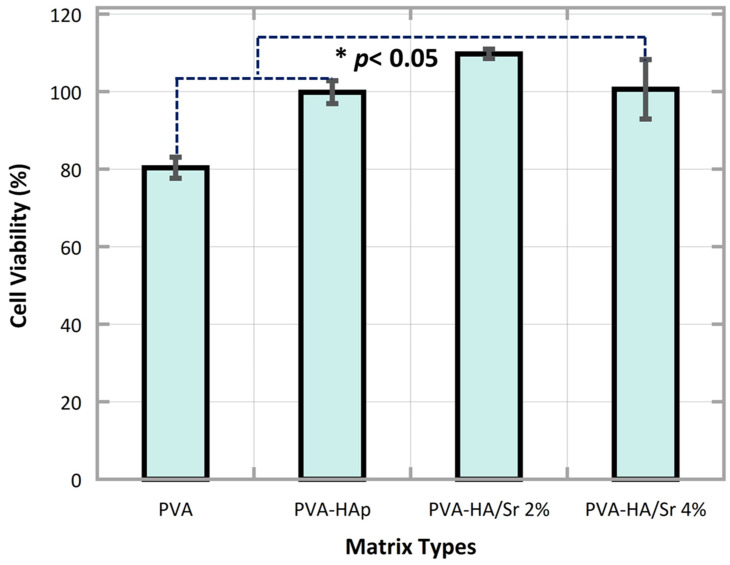
The results of MTT assay showed that PVA, PVA-HAp, PVA-HA/Sr-2 mol %, and PVA-HA/Sr-4 mol % fibrous matrices were found to be safe, nontoxic to the MC3T3-E1 cells. Cells viability was found higher than 80% in all matrices. The percentage of viable cells was >100 in PVA-HA/Sr-2 mol %. This indicated that proliferation started earlier in PVA-HA/Sr-2 mol %.

**Figure 10 bioengineering-10-00844-f010:**
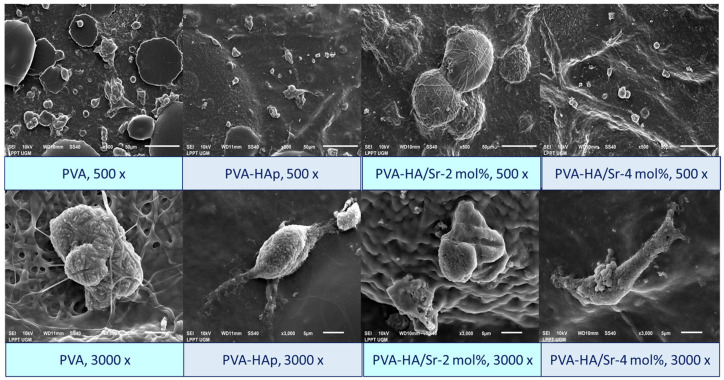
Morphology of the MC3T3-E1 cells on the surface of fibrous matrices. The cells were found to attach, extend their cytoplasms, and start to proliferate after 48 h of incubation. Observation was performed by SEM with 500 and 3000× magnification.

## Data Availability

All relevant data for this study can be obtained upon requested to the corresponding authors.

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
