# Peer review of "Fibrous PVA Matrix Containing Strontium-Substituted Hydroxyapatite Nanoparticles from Golden Apple Snail (Pomacea canaliculata L.) Shells for Bone Tissue Engineering"

_bioengineering, 2023, doi:10.3390/bioengineering10070844_

Round 1
Reviewer 1 Report
Poly(vinyl alcohol) or PVA could produce hydrogen bonds 67 to facilitate electrospinning process. Can the authors elucidate it more?
and strontium substituted hydroxyapatite (HA/Sr) synthesized-The authors suddenly introduced this concept of using strontium. May the authors please further clarify that why do they wanted to use HA/Sr rather than HAp and what benefits it can have?
Is this the first time that the HA/Sr combination is being used? Has other authors used this combination with PVA or other polymers before? If yes, please explain what is the novel aspect of this current work over those works?
A hydrothermal technique was used to synthesis-Typo error in this sentence and also other sentences. So please correct these typo/grammar errors.
Text in Fig. 1 is very small and unclear. Can the authors please look into this?
Next, diammonium hydrogen phosphate-Typo error in this formulae and also several others. So, please check all formulae carefully.
Please use a space between the number and unit, like temperature, volume, etc. Kindly check the whole paper including methods.
Thus, a ratio of 85:15 (v/v) 126 was used to combine the PVA solution and HA/Sr solution-Why this particular combination?
Please isnert formulae by using mathematical equations. The current one need to be modified, please.
Fig. 2 and Fig. 3 can be merged into 1 figure, please. Same is for Fig. 5 and Fig. 6. They can be combined.
become superior candidate to be applied in bone tissue engineering-As in this study, the exact potential of scaffolds for osteogenic TE has yet not been tested. So, this claim seems an axagerated.
Also about equation number, the authors may consider use numbering, such as Eq 1, Eq. 2, etc.
Before concluding, please enumerate limitations of this works. Such as biocompatibility evaluation using non-osteogenic cells, lack of complete profile of osteogenesis, lack of in vivo studies, etc.
Also, before conclusion, please clealry highlight take home message from this work. How the other researchers can use these materials and what are some of the expected implications from these results.
The text in both parts of Fig. 5 is unable to be read. So, please correct.
Please see the comments. There are substantial flawa in English Language, such as typo or grammar errors. Also the chemical formulae have errors. So, it needs an improvment, please.
Author Response
Dear Reviewer,
I attached here the response to the comment.

Reviewer 2 Report
The manuscript presents the characterization of nanofibrous matrix obtained using PVA and hydroxyapatite from golden snail shell. The manuscript can be considered for publication after adding the following:
- please specify in the material chapter what is SBF (cell culture media, physiological serum, serum, PBS)?
- add optical microscopy picture of MC3T3 cells cultivate in the presence of matrix
-can you add some elastic/resistance tests like young modulus
Minor editing of English language required.
Author Response
Dear Reviewer,
Thank you for the suggestions. I attach here the response for revision.
Best wishes

Reviewer 3 Report
The manuscript entitled “Nanofibrous PVA matrix containing strontium substituted hydroxyapatite nanoparticles from golden apple snail (Pomacea canaliculata L) shells for bone tissue engineering" was evaluated. This study reports the fabrication of a nanofibrous PVA-HA/Sr matrix made of strontium (Sr)-substituted hydroxyapatite from Pomecea canaliculate L. I think it could be considered for publication in “Bioengineering", meanwhile, some comments and minor corrections are needed as below:
Comment 1. Lines 17-20: Please be careful to provide the full names of the abbreviations (PVA, HA, Hap, and Sr) that have appeared for the first time in the abstract.
Comment 2. Lines 94, 96, 99, 110, 111, 118, 121, 126, 177, 187, 213: Please be careful to put a space before the “°C”.
Comment 3. The reference style of the Journal must be respected completely, e.g., “RSC Advances” in line 644 should be written as “RSC Adv”.
Comment 4. Line 409: Please provide more discussion regarding the SEM morphological analysis of the current study in comparison with the other studies.
Please check the text for some minor mistakes.
Author Response

(The authors gave the same response as above.)

Round 2
Reviewer 1 Report
The authors have kindly addressed the comments during revision. However, there are still some comments which need a further careful revision.
1. The text in Fig. 1 is too much and difficult to read
2. Scale bars on all SEM micrographs need to be manually added. The exisiting one can not be read, especially the text. Please address in Fig. 2, Fig. 4, Fig. 8.
3. The codes and text in Fig. 5 is of very small size. So, please increase the text size. In Fig. 5B, please label important peaks only.
4. There are many typo errors. For example in section 2.1. Materials. Also at several places there is no space between a number and a unit. Overall, the methods section has many typo errors
Also 360.50+77.16 to 505.51+78.60 nm; this and other such data
5. Please use an insert function to enter a formula. Kindly see equation 2.1. and 2.2. They have typo errors. Kindly express equation equation 2.1. by using symbols. The used symbols need description.
6. In 2.2.10. rather than using bullets, authors are encouraged to use numbering, such as 2.2.10.1., etc,
7. Rather than using the "nanofibrous", authors are kindly encouraged to use the term "fibrous" throughout the manuscript, please.
8. additional OH-1 functional groups...Please see how to accurately express it.
9. This reviewer still found some typo errors in the references. Such as Ref. 22.
The english editing of the manuscript is required, please.
Author Response
The authors have kindly addressed the comments during revision. However, there are still some comments which need a further careful revision.
- The text in Fig. 1 is too much and difficult to read
Thank you for the review and correction. We have revised Fig. 1 as suggested.
- Scale bars on all SEM micrographs need to be manually added. The existing one cannot be read, especially the text. Please address in Fig. 2, Fig. 4, Fig. 8.
We have added scale bars manually as suggested.
- The codes and text in Fig. 5 is of very small size. So, please increase the text size. In Fig. 5B, please label important peaks only.
We used the original graft from the XRD and FT-IR and we did not have access to TXT data to redraw the graph. That is why we can not follow the suggestion. However, to accommodate the good suggestion from reviewer, we modified graph caption.
- There are many typo errors. For example, in section 2.1. Materials. Also, at several places there is no space between a number and a unit. Overall, the methods section has many typo errors. Also 360.50+77.16 to 505.51+78.60 nm; this and other such data.
We have checked through the texts and revised the typos.
- Please use an insert function to enter a formula. Kindly see equation 2.1. and 2.2. They have typo errors. Kindly express equation equation 2.1. by using symbols. The used symbols need description.
We have followed the suggestions and used mathematical equation.
- In 2.2.10. rather than using bullets, authors are encouraged to use numbering, such as 2.2.10.1., etc,
We have revised the numbering as suggested.
- Rather than using the "nanofibrous", authors are kindly encouraged to use the term "fibrous" throughout the manuscript, please.
We have changed the use of nanofibrous into fibrous as suggested.
- additional OH-1 functional groups...Please see how to accurately express it.
We have checked and revised it.
- This reviewer still found some typo errors in the references. Such as Ref. 22.
We have checked and revised it.

Reviewer 2 Report
The manuscript in this form can be publish.
Author Response
The manuscript in this form can be publish.
Thank you very much for the review, comments, and suggestions.

Reviewer 3 Report
It appears that the authors have made the necessary correction, and it can be considered for publication.
The authors have made the necessary correction.
Author Response
It appears that the authors have made the necessary correction, and it can be considered for publication.
The authors have made the necessary correction.
Thank you very much for the review, comments, and suggestions.
